# High Expression of CUL9 Is Prognostic and Predictive for Adjuvant Chemotherapy in High-Risk Stage II and Stage III Colon Cancer

**DOI:** 10.3390/cancers14163843

**Published:** 2022-08-09

**Authors:** Peng Zheng, Yang Lv, Yihao Mao, Feifan Shen, Zhiyuan Zhang, Jiang Chang, Shanchao Yu, Meiling Ji, Qingyang Feng, Jianmin Xu

**Affiliations:** 1General Surgery Department, Zhongshan Hospital, Fudan University, 180 Fenglin Rd., Shanghai 200032, China; 2Shanghai Engineering Research Center of Colorectal Cancer Minimally Invasive, Shanghai 200032, China; 3Department of Surgery, Shanghai Medical College, Fudan University, Shanghai 200032, China

**Keywords:** CUL9, immunohistochemistry, colorectal cancer, adjuvant chemotherapy

## Abstract

**Simple Summary:**

This retrospective study evaluated the clinical implications of CUL9 expression on the prognosis and the predictive value for postoperative adjuvant chemotherapy in consecutive patients with colon cancer. Among all 1078 patients, high expression of CUL9 was an independent prognostic factor on disease-free survival and overall survival. Prognostic biomarkers are keys to the risk stratification of patients and the decision to recommend adjuvant chemotherapy. Among 564 patients with high-risk stage II and stage III disease who were recommended to receive the standard 6 months of chemotherapy, those with high CUL9 expression from the full dose group had better disease-free survival than those from the reduced dose group. A test for the interaction between CUL9 expression and the treatment reached significance and was not confounded by T stage, N stage and histopathological grade. This indicated that CUL9 expression may further filter those patients to identify truly high-risk cases that benefit from the full dose of the standard 6 months of chemotherapy.

**Abstract:**

We evaluated the clinical implications of CUL9 expression on the prognosis and the predictive value for adjuvant chemotherapy in colon cancer. A total of 1078 consecutive patients treated with radical resection from 2008 to 2012 were included. Formalin-fixed, paraffin-embedded specimens were used as immunohistochemistry (IHC) for CUL9. For all patients, high expression of CUL9 was identified as an independent prognostic factor for overall survival (HR = 1.613, 95% CI 1.305–1.993, *p* < 0.001) and disease-free survival (HR = 1.570, 95% CI 1.159–2.128, *p* = 0.004). The prognostic value of high CUL9 expression was confirmed in an independent validation cohort from the GEO database. The efficacy of adjuvant chemotherapy was analyzed among patients with high-risk stage II and stage III disease. Those with high CUL9 expression from the full dose group had better disease-free survival (HR = 0.477, 95% CI 0.276–0.825, *p* = 0.006) than those from the reduced dose group. The interaction test between CUL9 expression and the treatment reached significance and was not confounded by T stage, N stage and histopathological grade. In general, high expression of CUL9 was an independent prognostic factor in patients with colon cancer. In those with high-risk stage II and stage III disease, high expression of CUL9 was associated with the benefit from standard 6-months adjuvant chemotherapy regimens.

## 1. Introduction

Colorectal cancer (CRC) is one of the most common malignancies worldwide [1]. CRC is a heterogeneous disease with an overall 5-year survival rate below 60% [2]. There is an urgent need to identify patients with a high risk of recurrence and to improve the selection of patients who may benefit from adjuvant therapy or specific targeted therapy. Accordingly, stratification methods based on histopathological or molecular characteristics are extensively implemented, such as traditional clinical prognostic factors [3] (T4 tumor; tumor perforation; bowel obstruction; poorly differentiated tumor; venous invasion; <12 lymph nodes examined) and RAS mutation as a well-established determinant of resistance to anti-EGFR therapy [4,5]. However, current clinical and genetic profiling provides only a limited understanding of CRC. More potential markers, especially genetic alterations and their clinical implications are needed.

CUL9, also known as PARC (p53-associated Parkin-like cytoplasmic protein), belongs to the cullin family of proteins, which function as scaffolds with which to assemble E3 ubiquitin ligases [6]. Two unique features of CUL9 are that it localizes predominantly in the cytoplasm and binds to p53 [7,8]. To date, both the function and mechanism of CUL9 have not been fully defined. Accumulated evidence suggests that CUL9 plays a role in carcinogenesis and progression through a p53-dependent pathway [7,9,10,11,12].

As we previously reported, CUL9 may play a role in colorectal liver metastases (CLMs). Next-generation sequencing (NGS) was conducted in primary tumors of synchronous or metachronous CLMs. The results indicated that CUL9 mutation (rs2273709) was prevalent in CLMs [13] and had a predictive value for the efficacy of anti-EGFR therapy [14]. Consistent with our results, a study [15] conducted whole-exome sequencing (WES) for patients of all ages and cancer types. Mutational profiles of primary tumors and metastases yielded candidate mediators of metastatic spread, including CUL9. Therefore, we believe that further investigation is required to comprehensively understand the function and underlying mechanisms of CUL9 in CRC.

Because the expression of CUL9 is ubiquitous and high expression has been described in some tumor types [7], we further researched the clinical phenotype and function of CUL9 expression in colon cancer in this study.

## 2. Materials and Methods

### 2.1. Patients

In this study, two independent consecutive cohorts of patients were enrolled, including the primary cohort from a tissue microarray (TMA) and the validation cohort from the Gene Expression Omnibus (GEO) database.

We previously constructed a TMA of CRC from patients treated with radical resection at our center from June 2008 to December 2012 [16,17]. In this study, a total of 1078 consecutive patients with colon cancer were enrolled. Patients with rectal cancer that located less than 15 cm from anal verge according to preoperative colonoscopy were excluded. Clinical and pathological profiles were retrospectively reviewed. Pathological staging was determined according to the eighth edition of the UICC-TNM classification. The survival and recurrence status of survivors was last updated in 2020. Patients with high-risk stage II disease (characterized by at least one of the following: T4 tumor; tumor perforation; bowel obstruction; poorly differentiated tumor; venous invasion; <10 lymph nodes examined) were identified according to the NCCN guidelines. Informed consent was obtained from all patients for the acquisition and use of tissue samples and clinical data. This study was approved by the Clinical Research Ethics Committee of Zhongshan Hospital, Fudan University.

We also employed a patient set [18] from the GEO database for external validation. In this cohort, consecutive patients with stages I–IV colon cancer who underwent resections from 1987 to 2007 were collected from the French national Cartes d’Identité des Tumeurs program. All clinical and mRNA expression profiles were downloaded from the GEO website. Cases that were not colorectal tumors were excluded. Finally, 565 patients with colon cancer were enrolled in this study.

### 2.2. Immunohistochemistry (IHC)

TMA blocks were cut into 5-μm sections, de-paraffinized, rehydrated serially through alcohol and then stained and quantified. Briefly, slides were dried 2 h at 600 °C, the sections were dewaxed in xylene and graded alcohols, hydrated and washed in phosphate-buffered saline. After the endogenous peroxidase was inhibited by 3% H_2_O_2_ for 30 min, the sections were pretreated in a microwave oven (15 min in sodium citrate buffer, pH 6) and then incubated with 10% normal goat serum for 60 min. Primary antibodies recognizing CUL9 (diluted 1:40, HPA016434, Sigma-Aldrich, LLC., Darmstadt, Germany) were applied overnight in a moist chamber at 48 °C. Then the tissues were incubated with secondary antibodies (Sigma-Aldrich, LLC.), stained with diaminobenzidine (DAB) and counterstained with hematoxylin. The slides were scanned and quantified using Image Pro plus 6.0 (Media Cybernetics, Inc., Bethesda, MD, USA).

Two independent pathologists who were blinded to the clinical data evaluated the staining, and the results were averaged. Specifically, microscopic evaluation was performed within the tumor tissues in three randomly chosen fields of view. The area score was assessed (Stained area divided by total area per field: 0, less than 1%; 1, 1–25%; 2, 26–50%; 3, 51–75%; 4, 76–100%) and multiplied by an intensity score (0, none; 1, slight; 2, moderate; 3, strong staining) resulting in a total score range of 0–12 (IHC score). Quantitative results were expressed as averaged score of three fields per tumor tissue.

A previous study [19] demonstrated high concordance between the IHC-based mismatch repair (MMR) test and the PCR-based MSI test. In this study, MSI status was based on the IHC testing of MMR with four markers [19,20]: MLH1 (Clone G168-15, 1:50; BD Pharmingen, San Jose, CA, USA), MSH2 (Clone FE11, 1:200; Invitrogen, Waltham, MA, USA), MSH6 (Clone 44, 1:100; BD Pharmingen), and PMS2 (Clone A16-4, 1:100; BD Pharmingen). Patients with tumor tissue that exhibited positive staining for all these markers were considered proficient MMR (pMMR). Patients with negative staining for at least one marker were considered deficient MMR (dMMR).

### 2.3. Examination of RAS/BRAF Mutations

Formalin-fixed paraffin-embedded (FFPE) tissue was obtained from the Department of Pathology. An experienced pathologist reviewed each section and indicated the area of the tumor. Macro-dissection was performed using the H and E-stained slides to enrich the number of tumor cells in each sample. RAS/BRAF mutations were detected using the China Food and Drug Administration (CFDA) approved AmoyDx KRAS/NRAS/BRAF Mutations Detection Kit (AmoyDx, Xiamen, China), based on Amplification Refractory Mutation System (ARMS) technology in a certified laboratory. RAS and BRAF mutations examined were summarized in Appendix A.

### 2.4. Statistical Analysis

In both the primary and validation cohort, patients were divided into high and low expression groups according to the cutoff value of IHC score or mRNA expression of CUL9. The cutoff value was identified with X-tile software version 3.6.1 (Yale School of Medicine, New Haven, CT, USA) [6], representing the best prognostic efficacy on overall survival.

The correlations between CUL9 expression and the clinicopathological characteristics of the patients were analyzed using a chi-square test. Survival curves were estimated by the Kaplan–Meier method, and statistical significance was evaluated using a log-rank test. Hazard ratios (HRs) and 95% confidence intervals (95% CIs) were calculated using the Cox proportional hazards model. Overall survival (OS) was defined as the time from surgery to death, and disease-free survival (DFS) was defined as the time from surgery to disease recurrence. Univariate and multivariate analyses were performed using the Cox proportional hazards model. All statistical analyses were conducted using the statistical software SPSS version 25.0 (SPSS Inc., Chicago, IL, USA), and a *p* value < 0.05 was considered statistically significant.

## 3. Results

### 3.1. Baseline Characteristics and CUL9 Expression

The baseline characteristics are listed in Table 1. A total of 1078 consecutive patients with colon cancer were enrolled and divided into two groups according to CUL9 expression (Figure 1): a high expression group (*n* = 366) and a low expression group (*n* = 712). Baseline characteristics were analyzed between the groups. The results indicated that the high expression group was associated with positive lymph nodes, poor histological grade and more vascular/perineural invasion and tumor deposits (TDs). Genetic alterations, including RAS/BRAF mutations and MSI status, were not correlated with CUL9 expression.

### 3.2. Prognostic Value of CUL9 Expression

Compared with the low expression group, significantly worse OS was observed in patients from the high expression group (HR = 1.897, 95% CI 1.538–2.339, *p* < 0.001) (Figure 2A). Differences in OS between groups remained significant in stage II, III and IV patients, but not in stage I patients (Figure 2B–E).

In the univariate analysis, preoperative CEA level, histological grade, T stage, N stage, M stage, BRAF status, MSI status, vascular invasion, perineural invasion, TDs and CUL9 expression were considered potential prognostic variables of OS. In the multivariate analysis, high CUL9 expression was identified as an independent prognostic factor for OS (HR = 1.613, 95% CI 1.305–1.993, *p* < 0.001) (Table 2). In addition, TDs that showed prognostic value in the univariate analysis were not enrolled in the multivariate analysis because of the clear correlation with N stage.

In stage I–III patients (*n* = 785), the correlation between recurrence and CUL9 expression was analyzed. Significantly higher recurrence rates were observed in the high expression group (73/255, 28.6%) than in the low expression group (102/550, 18.5%) (OR = 1.762, 95% CI 1.246–2.491, *p* = 0.001). In the Kaplan–Meier analysis and log-rank test, the high expression group was associated with shorter DFS (HR = 1.596, 95% CI 1.182–2.156, *p* = 0.002) (Appendix A). Then, the prognostic value of high CUL9 expression for DFS was confirmed in the multivariate analysis (HR = 1.570, 95% CI 1.159–2.128, *p* = 0.004) (Appendix A).

### 3.3. Prognostic Value of CUL9 Expression in Validation Cohort

The validation cohort consisting of 565 consecutive patients with colon cancer was enrolled as a validation group. The baseline characteristics are listed in Appendix A. Consistent with the primary cohort, significantly worse OS was observed in patients from the high expression group (*n* = 183) than in those from the low expression group (*n* = 378) (HR = 1.360, 95% CI 1.015–1.824, *p* = 0.038) (Figure 2F). Simultaneously, in stage I–III patients, the high expression group was associated with a higher recurrence rate (35.6% vs. 24.9%, OR = 1.667, 95% CI 1.109–2.506, *p* = 0.014) and worse DFS (HR = 1.459, 95% CI 1.041–2.045, *p* = 0.027) (Appendix A).

In addition, the OS was analyzed according to the combination of CUL9 expression and p53 expression. Patients with high CUL9 expression and low p53 expression had the worst OS. The OS of patients with other combinations of CUL9 and p53 expression were comparable (Appendix A).

### 3.4. Predictive Value of CUL9 Expression for Efficacy of Adjuvant Chemotherapy

In the primary cohort, 263 patients were identified as having high-risk stage II disease and 301 patients as having stage III disease. Among all patients with high-risk stage II and stage III disease (*n* = 564), significantly higher recurrence rates were observed in the high expression group (61/180, 33.9%) than in the low expression group (84/384, 21.9%) (OR = 1.831, 95% CI 1.237–2.710, *p* = 0.003). The high expression group was also associated with worse DFS (HR = 1.629, 95% CI 1.171–2.265, *p* = 0.004). (Figure 3A)

High-risk stage II and stage III patients were recommended to receive the standard 6 months of adjuvant chemotherapy according to guidelines. In this study, all those patients were divided into three groups according to finished cycles of chemotherapy regimens: full dose group (at least 75% of planned cycles, *n* = 364), reduced dose group (25% to 75% of planned cycles, *n* = 129) and low dose group (at most 25% of planned cycles, *n* = 51) (Appendix A page 2). According to the Kaplan–Meier curves, better DFS was observed in the full dose group than in the reduced dose group, but the difference did not reach statistical significance (*p* = 0.106). In the high expression group, patients from the full dose group had better DFS (HR = 0.477, 95% CI 0.276–0.825, *p* = 0.006) than those from the reduced dose group. A test for the interaction between CUL9 expression and the treatment revealed that the benefit observed in the high expression group was superior to that observed in the low expression group (Figure 3B–D; Appendix A). This effect was not confounded by many known risk factors, including tumor location, T stage, N stage, histopathological grade, perineural invasion and BRAF status (*p* = 0.019). These results indicated that CUL9 expression may be predictive of adjuvant chemotherapy benefit in high-risk stage II and stage III patients.

Subgroup analysis according to clinicopathological variables indicated that patients with T4 tumor from the full dose group had better DFS (HR = 0.545, 95% CI 0.312–0.950, *p* = 0.032) than those from the reduced dose group (Figure 4). The interaction test failed to reach statistical significance (*p* = 0.180).

## 4. Discussion

Prognostic biomarkers are keys to the risk stratification of patients with CRC and the decision to recommend adjuvant chemotherapy. Currently, histopathological characteristics, such as tumor stage, remain the most important among a handful of prognostic variables. Meanwhile, molecular characteristics have been noted. For example, CDX2 was identified through a bioinformatics approach and proved to be prognostic and predictive for the efficacy of adjuvant chemotherapy in patients with stage II and III colon cancer [21]. In our previous study, an association between CUL9 and CLMs was observed through NGS [13]. CUL9 is the largest and youngest member of the cullin family and has many domains [22]. To our knowledge, the clinical phenotype and function of CUL9 have not been described to date for colon cancer. In this study, we showed that high expression of CUL9 is an independent prognostic factor for OS and DFS in patients with colon cancer. Furthermore, it may be predictive of recurrence and the efficacy of the standard 6 months of adjuvant chemotherapy in high-risk stage II and stage III patients.

In our results, the prognostic value of CUL9 expression was further confirmed in the external validation cohort that consisted of consecutive patients with colon cancer from France. Moreover, prognostic value was also observed in a dataset consisting of 597 patients with CRC from the Cancer Genome Atlas (TCGA) database (Appendix A). Similar results across different cohorts indicated that CUL9 plays a role in CRC. To date, both the function and mechanism of CUL9 have not been fully defined. According to the current understanding, CUL9 has been validated to function as a cytoplasmic anchor for the p53 protein. Inactivation of CUL9 promotes nuclear localization of p53 and triggers cell apoptosis, while overexpression of CUL9 induces cytoplasmic sequestration of p53 [7]. Indeed, constitutive cytoplasmic localization of p53 has been linked to poor response to chemotherapy, tumor metastasis, and poor long-term patient survival in many tumor types, including colorectal carcinoma [23,24,25,26]. Notably, in the validation cohort, patients with high CUL9 expression and low p53 expression had the worst OS. The OS of patients with other combinations of CUL9 and p53 expression were comparable (Appendix A). This may add evidence to the hypothesis that CUL9 plays a role in CRC through the p53-dependent pathway. Furthermore, we tried to reveal the mechanism of CUL9 in CRC cell lines. The results which were published recently indicated that CUL9 can bind p53 to ubiquitylate heterogeneous nuclear ribonucleoprotein C for degradation. CUL9 is a ferroptosis response modulator in CRC, and it is mediated by the CUL9-HNRNPC-MATE1 negative loop [27].

Our results also indicated that patients with high CUL9 expression have a higher risk of recurrence or diagnosis of stage IV disease. As we previously reported, WES was conducted for 10 triplets, each comprising primary colorectal tumor and normal colorectal mucosa and matched liver metastases, and 96 genes essential for cancer progression were screened out. Then, NGS was performed in 93 synchronous and 68 metachronous CLMs. CUL9 mutation (rs2273709) was prevalent in two types of CLMs with similar frequency [13]. Thus, CUL9 likely has important clinical implications for the prediction of occurrence, despite the timing of metastases. Pisapia et al. [15] conducted WES for patients of all ages and cancer types. The mutational profile of the primary tumor and metastases from a patient with recurrent anaplastic ependymoma yielded candidate mediators of metastatic spread, including CUL9 and PIGM. More sequencing data of primary tumors and metastases would permit the identification of the role of CUL9 in tumor progression, including whether it contributes to metastatic potential or even treatment resistance.

Typically, postoperative standard 6 months of adjuvant chemotherapy is recommended for patients with high-risk stage II and stage III CRC. However, our results suggested that CUL9 expression can further filter these patients to identify truly high-risk cases. Although the difference of DFS between the full dose group and the reduced dose group did not reach significance, those patients with high CUL9 expression from the reduced dose group had much worse DFS and were supposed to receive the full dose of standard 6 months of chemotherapy regimens. We hypothesize that the predictive value associated with CUL9 expression could be partly explained by its prognostic value. In addition, CUL9 probably had a direct effect on oxaliplatin resistance, as CUL9 regulates p53 subcellular localization and apoptosis, leading to an effect on cisplatin resistance in ovarian cancer cells [28]. Recently, in the context of precision medicine, the duration of adjuvant chemotherapy has been widely discussed. The critical IDEA trial [29] evaluated the noninferiority of 3 months compared with the standard 6 months of adjuvant fluoropyrimidine plus oxaliplatin in patients with stage III colon cancer. Post hoc analysis of patients with low-risk (T1–3N1) cases revealed that 3 months of therapy was noninferior to 6 months, and among those classified as high risk (T4, N2 or both), 6 months of therapy was superior to 3 months [30]. Similar with the IDEA trial, our results also indicated that full dose regimens were superior to reduced dose among patients with T4 tumor. All the above results together suggested that adjuvant chemotherapy should be administered in a more precise manner: the higher risk of recurrence, the more intense chemotherapy regimens. Accordingly, more candidate variables and a stratification system are needed.

Our study had several limitations. First, inherent to any TMA study is the possibility of selection bias. Therefore, we excluded necrotic and fibrotic cores before creating the TMA and collected averaged results from two pathologists. Second, patient selection bias and recall bias were possibilities due to the nature of retrospective studies. Therefore, we enrolled consecutive patients who received surgical resections and employed validation cohorts from independent centers. Third, the results of the IDEA trial [29,30] showed a difference in the performance of the CAPOX and FOLFOX regimens. We did not perform subgroup analysis by regimens because of the limited sample size.

## 5. Conclusions

Our results indicated that high CUL9 expression was associated with worse OS in patients with colon cancer. In those with high-risk stage II and stage III disease, high CUL9 expression was associated with a higher rate of recurrence and benefit from the standard 6 months of adjuvant chemotherapy. Given the exploratory and retrospective design of our study, these results need to be further validated.

## Figures and Tables

**Figure 1 cancers-14-03843-f001:**
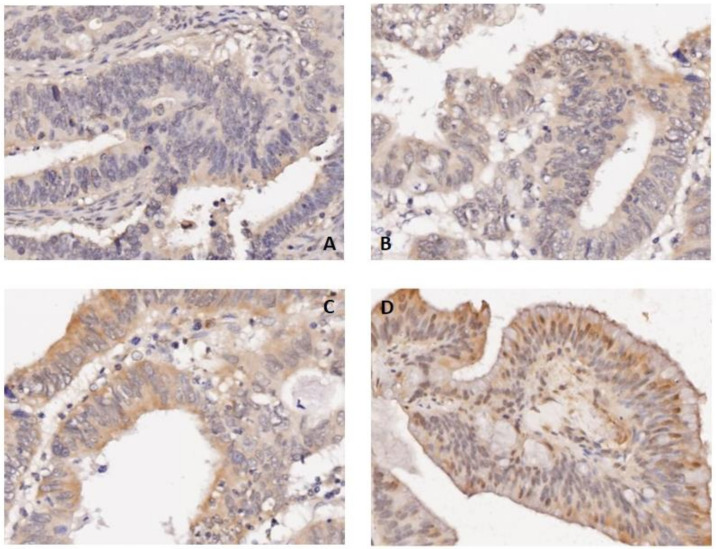
Representative images of IHC staining of CUL9. Tumor tissue with staining intensity from negative to strong positive (**A**–**D**) is presented successively. Magnification of all images, ×20.

**Figure 2 cancers-14-03843-f002:**
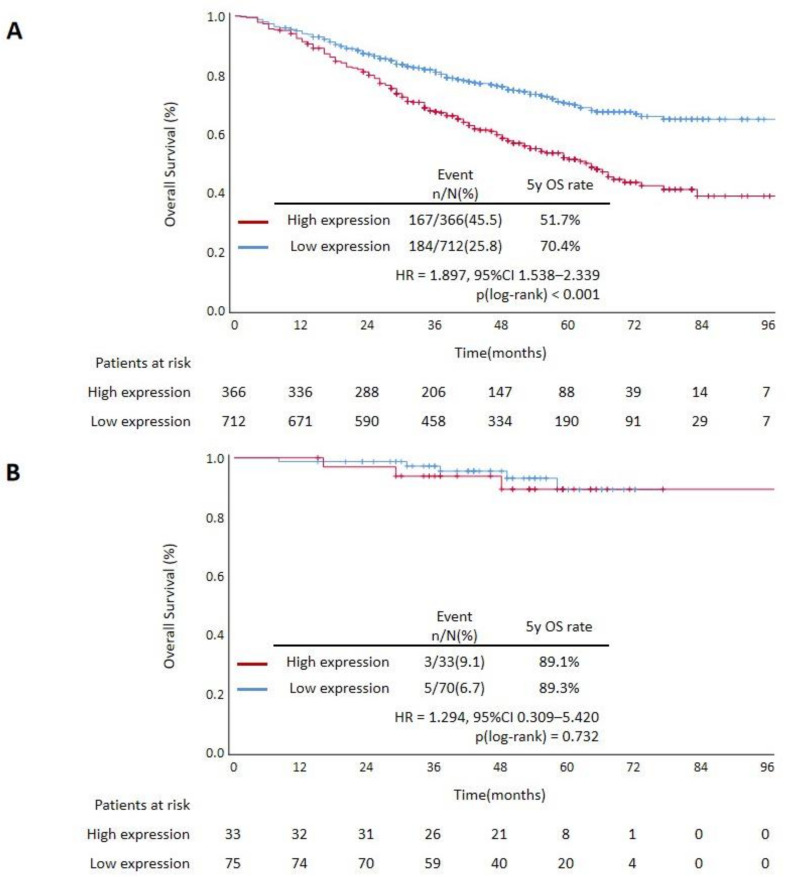
Kaplan–Meier curves of OS stratified by CUL9 expression: (**A**) all patients; (**B**) stage I patients; (**C**) stage II patients; (**D**) stage III patients; (**E**) stage IV patients; (**F**) all patients in the validation cohort; HR, hazard ratio; CI, confidence interval.

**Figure 3 cancers-14-03843-f003:**
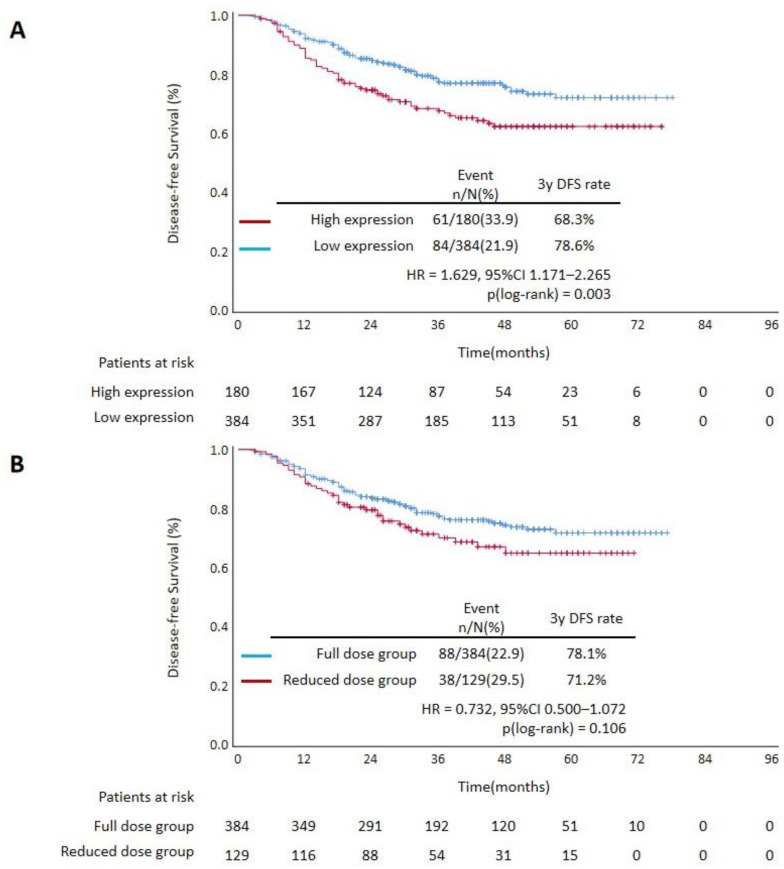
Kaplan-Meier curves of DFS stratified by adjuvant chemotherapy in high-risk stage II and stage III patients: (**A**) high CUL9 expression group vs. low expression group; (**B**) full dose group vs. low douse group among all patients; (**C**) full dose group vs. low dose group among patients with high CUL9 expression; (**D**) full dose group vs. low dose group among patients with low CUL9 expression. HR, hazard ratio; CI, confidence interval.

**Figure 4 cancers-14-03843-f004:**
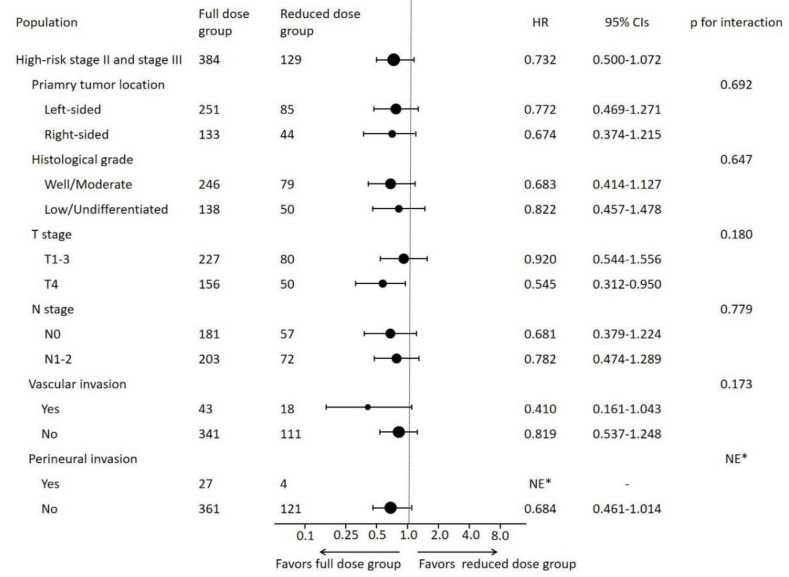
Forest plots of DFS stratified by adjuvant chemotherapy in high-risk stage II and stage III patients. Subgroups were divided according to traditional clinicopathological high-risk variables: HR, hazard ratio; CI, confidence interval, NE, not evaluable. * There were no DFS events in subgroup from reduced dose group with perineural invasion.

**Table 1 cancers-14-03843-t001:** Correlation between baseline characteristics and CUL9 expression in the primary cohort.

	Total (*n* = 1078)	High Expression Group (*n* = 366)	Low Expression Group (*n* = 712)	*p* Value
Age, years, n (%)				0.835
≥60	602 (55.8%)	206 (56.3%)	396 (55.6%)	
<60	476 (44.2%)	160 (43.7%)	316 (44.4%)	
Gender, n (%)				0.965
Male	647 (60.0%)	220 (60.1%)	427 (60.0%)	
Female	431 (40.0%)	146 (39.9%)	285 (40.0%)	
Pre-operative CEA, ng/mL, n (%)				0.049
>5	500 (46.4%)	185 (50.5%)	315 (44.2%)	
≤5	578 (53.6%)	181 (49.5%)	397 (55.8%)	
Mean tumor size, cm, ±SD	4.28 ± 2.03	4.37 ± 2.13	4.23 ± 2.05	0.541
Tumor location, n (%)				0.760
Right-sided	363 (33.7%)	121 (33.1%)	242 (34.0%)	
Left sided	715 (66.3%)	245 (66.9%)	470 (66.0%)	
Histological grade, n (%)				0.041
Well/Moderate	730 (67.7%)	233 (63.7%)	497 (69.8%)	
Low/Undifferentiated	348 (32.3%)	133 (36.3%)	215 (30.2%)	
T stage, n (%)				0.321
T1–T2	149 (13.8%)	43 (11.7%)	106 (14.9%)	
T3	494 (48.9%)	168 (45.9%)	326 (45.8%)	
T4	435 (40.4%)	155 (42.3%)	280 (39.3%)	
N stage, n (%)				0.010
N0	577 (53.5%)	176 (48.1%)	401 (56.3%)	
N1–2	501 (46.5%)	190 (51.9%)	311 (43.7%)	
TNM stage, n (%)				0.039
I	108 (10.0%)	33 (9.0%)	75 (10.5%)	
II	396 (36.7%)	120 (32.8%)	276 (38.8%)	
III	301 (27.9%)	102 (27.9%)	199 (27.9%)	
IV	273 (25.3%)	111 (30.3%)	162 (22.8%)	
Vascular invasion, n (%)				0.025
Yes	134 (12.4%)	57 (15.6%)	77 (10.8%)	
No	944 (87.6%)	309 (84.4%)	635 (89.2%)	
Perineural invasion, n (%)				0.016
Yes	85 (7.9%)	39 (10.7%)	46 (6.5%)	
No	993 (92.1%)	327 (89.3%)	666 (93.5%)	
Tumor deposits, n (%)				<0.001
Yes	245 (22.7%)	108 (29.5%)	137 (19.2%)	
No	833 (77.3%)	258 (70.5%)	575 (80.8%)	
RAS * status, n (%)				0.179
Wild-type	499 (46.3%)	159 (43.4%)	340 (47.8%)	
Mutant	579 (53.7%)	207 (56.6%)	372 (52.2%)	
BRAF V600E status, n (%)				0.560
Wild-type	1008 (93.5%)	340 (92.9%)	668 (93.8%)	
Mutant	70 (6.5%)	26 (7.1%)	44 (6.2%)	
MMR status, n (%)				0.888
pMMR	973 (90.3%)	331 (90.4%)	642 (90.2%)	
dMMR	105 (9.7%)	35 (9.6%)	70 (9.8%)	

Abbreviation: CEA, carcinoembryonic antigen; MMR, mismatch repair; pMMR, proficient mismatch repair; dMMR, deficient mismatch repair. *: Profile of RAS mutation loci was summarized in Appendix A.

**Table 2 cancers-14-03843-t002:** Uni- and multivariate analysis of overall survival in the primary cohort.

	Univariate Analysis			Multivariate Analysis *		
	HR	95% CI	*p* Value	HR	95% CI	*p* Value
Age, years			0.375			
<60	1	-				
≥60	0.909	0.737–1.122				
Gender			0.480			
Female	1	-				
Male	1.081	0.871–1.341				
Pre-operative CEA, ng/mL			<0.001			0.088
≤5	1	-		1	-	
>5	2.596	2.086–3.230		1.232	0.969–1.567	
Tumor location			0.104			
Right-sided	1	-				
Left-sided	0.835	0.672–1.037				
Histological grade			<0.001			0.338
Low/Undifferentiated	1	-		1	-	
Well/Moderate	0.613	0.495–0.759		0.896	0.715–1.122	
T stage						
T1–2	1	-		1	-	
T3	2.141	1.382–3.316	0.001	1.005	0.638–1.586	0.981
T4	2.992	1.936–4.624	<0.001	1.105	0.696–1.752	0.673
N stage						
N0	1	-		1	-	
N1	2.474	1.927–3.176	<0.001	1.533	1.183–1.987	0.001
N2	4.689	3.577–6.146	<0.001	2.319	1.722–3.123	<0.001
M stage			<0.001			<0.001
M0	1	-		1	-	
M1	9.401	7.531–11.736		6.818	5.303–8.765	
Vascular invasion			<0.001			0.063
No	1	-		1	-	
Yes	2.237	1.715–2.917		1.314	0.986–1.751	
Perineural invasion			<0.001			0.581
No	1	-		1	-	
Yes	2.261	1.636–3.123		1.103	0.780–1.558	
Tumor deposits ^†^			<0.001			
No	1	-				
Yes	2.680	2.161–3.323				
RAS status			0.603			
Wild-type	1	-				
Mutant	1.139	0.922–1.406				
BRAF status			<0.001			<0.001
Wild-type	1	-		1	-	
Mutant	2.389	1.741–3.278		1.859	1.343–2.574	
MMR status			0.044			0.036
dMMR	1	-		1	-	
pMMR	0.658	0.438–0.988		0.644	0.427–0.971	
CUL9 expression			<0.001			<0.001
Low	1	-				
High	1.897	1.538–2.339		1.613	1.305–1.993	

Abbreviation: CEA, carcinoembryonic antigen; MMR, mismatch repair; pMMR, proficient mismatch repair; dMMR, deficient mismatch repair; HR, hazard ratio; CI, confidence interval. *: Multivariate analysis included those variates with *p* < 0.10 in univariate analysis. ^†^: N1c stage was defined as that no regional lymph nodes are positive, but tumor deposits are detected. Therefore, tumor deposit was not included in multivariate analysis because of the clear correlation with N stage.

## Data Availability

The data of the primary cohort presented in this study are available on request from the corresponding author. The data are not publicly available due to conditions of the ethics committee of our center. The data of the validation cohorts are publicly available on the following websites. GEO: https://www.ncbi.nlm.nih.gov/geo/query/acc.cgi?acc=GSE39582 (accessed on 29 April 2022). TCGA: https://www.proteinatlas.org/ENSG00000112659-CUL9/pathology/colorectal+cancer (accessed on 3 May 2022).

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
