# Peer review of "High Expression of CUL9 Is Prognostic and Predictive for Adjuvant Chemotherapy in High-Risk Stage II and Stage III Colon Cancer"

_cancers, 2022, doi:10.3390/cancers14163843_

Round 1

Reviewer 1 Report

In my opinion, it's a well organized study now. No comment from me at this time.

Reviewer 2 Report

The manuscript has been improved by the authors however more information/data would be have to be provided by the authors to consider CUL9 as a prognostic and predictive marker in II and III stages of colon cancer.

Reviewer 3 Report

All correct.

This manuscript is a resubmission of an earlier submission. The following is a list of the peer review reports and author responses from that submission.

Round 1

Reviewer 1 Report

This retrospective study evaluated the clinical implications of CUL9 expression 19 on prognosis and the predictive value for postoperative adjuvant chemotherapy in consecutive patients with colon cancer. It’s a well-written and rigorous study.

Comment 1

What’s the criterion of high expression and low expression? I didn’t find it in the article.

Comment 2

Are there any exclusion criteria in your study?

Comment 3

In line 234-237, how did the author prove the effect was not confounded by other factor?

Reviewer 2 Report

The manuscript by Zheng et al., is a description of CUL9 as a novel prognostic marker and a predictive marker for adjuvant therapy treatments by using a cohort of 1,078 patients.

The study is well designed, presents novelty and has a potential clinical translation. The authors show the relevance of CUL9 in colon cancer by doing immunohistochemistry (TMAs) and analysing patient's data (survival and so on). They discuss how CUL9 would be acting, however, there are no insights about CUL9 mechanism of action in the results section.  The manuscript would need more information related to CUL9 pathways and a proof experimentally that it is a relevant factor in the disease.

Reviewer 3 Report

In this manuscript, Zheng et al evaluate the clinical implications of CUL9 y colon cancer cohorts. The study contains some interesting observations. Yet, data sometimes is confusing:

1/ Authors should review the number of patients along the manuscript, as sometimes it does not match the figure (for example 263 patients identified as having high-risk stage II disease but in Figure 2C is indicated otherwise).

2/ Misspelling of Figure 2F (is not 2E)

3/ Instead of "Survival curves of DFS..." in the legend, "Kaplan-Meier curves of DFS..." should be written as DFS is not survival.

4/ There is no legend for Supplementary Figures S1 and S2. 

5/ Authors described a possible correlation between CUL9 and p53 in the discussion but this should be in the Results section. Moreover, the Supplementary Figure S3 related to this part is not in the supplementary material. 

6/ Authors claim in the discussion that "Moreover, prognostic value was also observed in a dataset consisting of 597 patients with CRC from the TCGA database" but this is lacking in the manuscript. 

Reviewer 4 Report

1.There is a description about adjuvant chemotherapy on line223-237. It is divided into 3 groups: Full dose, reduced dose and low dose. Are the three categories only for the number of courses, or are there any considerations such as dose reduction? Also, are all adjuvant chemotherapy a combination of fluoropyrimidines and oxaliplatin?

2.line280-287, you mentioned that CUL9 is  linked poor response to chemotherapy. But after looking at the 23,24 papers you cited, there are few references to the link between p53 and the effects of chemotherapy. You also mention that adjuvant chemotherapy is effective in CUL9 overexpression cases in this study. CUL9 overexpression may not be a predictor of the effect of chemotherapy, but a predictor of prognosis.

3.There are parts with different fonts in multiple places, so please correct them.